# Beneficial Effects of Capsaicin in Disorders of the Central Nervous System

**DOI:** 10.3390/molecules27082484

**Published:** 2022-04-12

**Authors:** Michał Pasierski, Bartłomiej Szulczyk

**Affiliations:** Department of Pharmacodynamics, The Medical University of Warsaw, Banacha 1B, 02-097 Warsaw, Poland; michalpasierski@gmail.com

**Keywords:** capsaicin, neurodegenerative diseases, stroke, epilepsy, depression

## Abstract

Capsaicin is a natural compound found in chili peppers and is used in the diet of many countries. The important mechanism of action of capsaicin is its influence on TRPV1 channels in nociceptive sensory neurons. Furthermore, the beneficial effects of capsaicin in cardiovascular and oncological disorders have been described. Many recent publications show the positive effects of capsaicin in animal models of brain disorders. In Alzheimer’s disease, capsaicin reduces neurodegeneration and memory impairment. The beneficial effects of capsaicin in Parkinson’s disease and depression have also been described. It has been found that capsaicin reduces the area of infarction and improves neurological outcomes in animal models of stroke. However, both proepileptic and antiepileptic effects of capsaicin in animal models of epilepsy have been proposed. These contradictory results may be caused by the fact that capsaicin influences not only TRPV1 channels but also different molecular targets such as voltage-gated sodium channels. Human studies show that capsaicin may be helpful in treating stroke complications such as dysphagia. Additionally, this compound exerts pain-relieving effects in migraine and cluster headaches. The purpose of this review is to discuss the mechanisms of the beneficial effects of capsaicin in disorders of the central nervous system.

## 1. Introduction

Capsaicin is a natural compound, often used in many countries during food preparation, responsible for the spiciness of chili peppers. Capsaicin belongs to a group of compounds called vanilloids [1,2]. The widely-described mechanism of action of capsaicin is the activation of calcium-permeable TRPV1 channels. These receptors are also activated by other stimuli, for example, noxious temperature, acidic pH, and endogenous chemical activators such as endocannabinoids [1]. Capsaicin exerts its effects on different organs because TRPV1 channels are expressed in different cell types, for example, neuronal, muscular, immune, and epithelial cells as well as adipocytes [1]. Moreover, other molecular targets for capsaicin such as voltage-gated sodium and calcium channels have been described [3,4].

Capsaicin exerts beneficial effects in many disorders [2]. It was reported, for example, that capsaicin exerts anti-obesity effects by changing gut-microbiota compositions [5]. Moreover, capsaicin induces apoptosis in different types of cancer cell lines, as shown by in vitro experiments. In vivo studies have shown that capsaicin reduces the growth of many tumors in animal models of oncological disorders [6]. In the vascular system, capsaicin stimulates TRPV1 receptors expressed on endothelial cells and increases the production of vasodilating factors [7]. This process reduces ischemic injury which contributes to the pathogenesis of cardiovascular and neurological disorders [7,8].

In the peripheral nervous system, capsaicin acts on pain receptors (nociceptors) in the skin and mucosa and evokes burning sensations by opening TRPV1 channels. Surprisingly, capsaicin also exerts analgesic effects and, consequently, this compound is used in treating neuropathic pain. It has been suggested that capsaicin relieves pain by causing desensitization of TRPV1 channels expressed in peripheral pain receptors [9]. Moreover, capsaicin inhibits voltage-gated sodium currents which reduces the excessive activity of sensory neurons and contributes to the pain-relieving effects of this compound [10]. Capsaicin also influences brain neurons and glial cells via TRPV1-dependent and independent mechanisms. Many different positive effects of capsaicin in neurodegenerative diseases, epilepsy, stroke, and depression have been described in animal and human studies [3,11,12,13,14]. The aim of this review is to describe the beneficial effects of capsaicin in disorders of the central nervous system.

## 2. Physico-Chemical and Pharmacokinetic Properties of Capsaicin

Capsaicin is an odorless, colorless, crystalline compound (its chemical formula is represented in Figure 1). It has a melting point of about 63 Degrees Celsius and its molar mass is 305.4 g/mol. It is not soluble in water; however, it is well soluble in organic solvents such as ethanol or DMSO [15].

Animal studies show that the absorption of capsaicin after oral intake varies from 50 to 90% [15]. In one study conducted on humans, after oral administration of 5 g of chili peppers (which amounted to 26.6 mg of pure capsaicin), the peak plasma concentration was 2.47 ± 0.13 ng/mL, time to peak concentration was 47.1 ± 2.0 min and t_1/2_ was 24.9 ± 5.0 min [16].

Capsaicin crosses the blood–brain barrier efficiently, as shown by animal studies. After intravenous administration, 5-fold higher concentrations of the compound in the brain and spinal cord were reported compared with serum [17]. Additionally, it was shown in animal studies that capsaicin could be detected in the plasma, brain, and spinal cord after subcutaneous administration [15].

## 3. Alzheimer’s Disease

Alzheimer’s disease is a common neurodegenerative disease. Its main histopathological features are beta-amyloid accumulation, tau hyperphosphorylation, and the death of brain cells. The main symptoms of this disease are memory loss and psychiatric disturbances [18]. Certain drugs such as cholinesterase inhibitors are administered to slow the progress of this disorder [19]. Moreover, a proper diet may be beneficial in Alzheimer’s disease because healthy food contributes to cognitive health during aging [20]. Transgenic mice expressing a mutated human amyloid precursor protein are often used as a model of Alzheimer’s disease [18].

### 3.1. Beneficial Effects of Capsaicin in Animal Models of Alzheimer’s Disease

Microglia are immune cells of the central nervous system. An important function of these cells is phagocytosis of the beta-amyloid plaques in the early phases of Alzheimer’s disease. This function is then impaired in more advanced diseases [21,22,23]. Autophagy is a process of intracellular degradation of organelles and toxic molecules which is involved in the aging process [23]. It was reported that treatment of microglial cultures with capsaicin (10 µM) induced autophagy and increased the phagocytic capacity of microglial cells. Consequently, beta-amyloid was more efficiently removed from the culture medium. TRPV1 receptors expressed in microglia were involved in this effect. It was shown in the same study that intraperitoneal administration of capsaicin (1 mg/kg daily for one month) reduced beta-amyloid deposition and improved learning and memory in transgenic mice with Alzheimer’s disease. The authors concluded that capsaicin ameliorated symptoms of Alzheimer’s disease by inducing autophagy in microglial cells and by increasing their phagocytic capacity [23]. In a different study, it was found that the metabolic and phagocytic function of microglia was impaired by beta-amyloid. The authors demonstrated that this impairment was reduced after treatment of microglia cultures with capsaicin (10 µM). Thus, the phagocytosis of beta-amyloid by microglial cells was more efficient after treatment with capsaicin. This beneficial effect was dependent on TRPV1 channels. The same authors, in behavioral experiments, found that dietary capsaicin (0.01% in a chow) improved memory in a mouse model of Alzheimer’s disease [21].

It was found in mice with Alzheimer’s disease, that dietary capsaicin (0.01% in a chow) reduced beta-amyloid plaque formation and tau phosphorylation in different brain areas, and attenuated neurodegeneration and cognitive impairment [23]. Beta-amyloid is a protein produced by enzymatic cleavage of the amyloid precursor protein. Cited authors provided evidence that in capsaicin-fed mice with Alzheimer’s disease, an enzyme alpha-secretase was up-regulated which cleaved amyloid precursor protein within the beta-amyloid domain. This reduced beta-amyloid production [23]. The beneficial effects of capsaicin in Alzheimer’s disease are shown in Figure 2 in simplified form.

Long-term potentiation (LTP) is the persistent increase in synaptic strength evoked by high-frequency synaptic stimulation. In other words, the rising slope and/or maximal amplitude of excitatory postsynaptic potential is increased after high-frequency synaptic stimulation. This phenomenon is often recorded in hippocampal and cortical slices obtained from animals. Long-term potentiation (LTP) is the electrophysiological correlate of certain forms of memory. Consequently, LTP is impaired in Alzheimer’s disease [24,25].

In one study, pathological changes resembling Alzheimer’s disease were induced by intracerebral microinjection of beta-amyloid in mice. Electrophysiological experiments showed impaired hippocampal long-term potentiation (LTP) whereas behavioral experiments revealed impaired spatial memory and learning. These alterations were accompanied by a decreased number of synapses in beta-amyloid-treated mice as assessed by electron microscopy techniques. The authors reported that intraperitoneal capsaicin (1 mg/kg) improved spatial memory, enhanced LTP, and reduced synapse loss in tested animals [24].

In a different publication, it was found that intraperitoneal administration of TRPV1 agonist capsaicin (1 mg/kg) reduced spatial learning and memory impairments in a mouse model of Alzheimer’s disease [25]. The authors also reported that genetic upregulation of TRPV1 receptors reduced beta-amyloid deposition in the same model of Alzheimer’s disease. Similarly, to the study by Chen and colleagues [24], histological and behavioral experiments were confirmed by electrophysiological studies. The authors found that genetic upregulation of TRPV1 receptors ameliorated hippocampal long-term potentiation (LTP) impairment in mice with Alzheimer’s disease. The authors also presented evidence that TRPV1 upregulation reduced LTP impairment by inhibiting AMPA receptor endocytosis [25].

Neuronal network gamma oscillations are correlated with cognitive processes. These EEG patterns decrease in Alzheimer’s disease patients [26]. An interesting electrophysiological study showed protective effects of capsaicin (10 μM) against neuronal gamma oscillations impairment induced by beta-amyloid. Gamma oscillations were recorded using extracellular recordings in hippocampal slices. The authors found that the application of beta-amyloid strongly reduced the amplitude of gamma oscillations and that this reduction was almost abolished in the presence of capsaicin. This positive effect of capsaicin was absent in TRPV1 knock-out mice. Additionally, the co-application of TPV1 antagonist with capsaicin did not protect gamma oscillations from beta-amyloid-induced impairment. Thus, the effect of capsaicin depended on TRPV1 channels [27].

One publication has suggested that capsaicin may deteriorate symptoms of Alzheimer’s disease. In this study, the level of beta-amyloid was assessed in cell lines expressing amyloid precursor protein. It was found that the level of beta-amyloid increased after preincubation with capsaicin (10 μM). Moreover, capsaicin impaired the degradation of beta-amyloid [28]. The authors suggested the limitation of their study was that tested cells did not express TRPV1 channels and were incubated with a high concentration of capsaicin for quite a short period of time (24 h). On the other hand, beneficial dietary capsaicin intake may last many years and provides exposure to low concentrations of capsaicin. This may be an explanation for why the authors [28] presented different conclusions than many other studies, which suggested beneficial effects of capsaicin in Alzheimer’s disease (see above).

### 3.2. Capsaicin Reduces the Risk of Alzheimer’s Disease

It was also reported in both human and animal subjects that capsaicin reduces the risk of Alzheimer’s disease. In western regions of China, chili peppers are more often consumed and there is a smaller number of people with dementia than in other regions where dietary capsaicin intake is lower [29]. It was found in healthy humans that a capsaicin-rich diet correlates with reduced levels of serum beta-amyloid which indicates a lower risk of Alzheimer’s disease. In the same subjects, it was also found that a capsaicin-rich diet improved cognition [29]. In a different study, it was shown that dietary capsaicin (0.01% in a chow) reduced the risk of Alzheimer’s disease in rats with type 2 diabetes, which is a risk factor for Alzheimer’s disease. One of the mechanisms of this effect was that the capsaicin-rich diet reduced tau protein phosphorylation in the hippocampus of diabetic rats. Additionally, the authors found that dietary capsaicin decreased blood glucose concentration in diabetic rats which suggests that this compound may be beneficial in treating diabetes [30].

The publications presented above combine to suggest that capsaicin exerts beneficial effects in animal models of Alzheimer’s disease. Moreover, capsaicin intake may reduce the risk of Alzheimer’s disease. Some of the mechanisms are presented in Figure 2 in simplified form.

## 4. Beneficial Effects of Capsaicin in Parkinson’s Disease

The main symptoms of Parkinson’s disease are tremor, rigidity, and bradykinesia, all of which are caused by the degeneration of tyrosine hydroxylase positive dopaminergic neurons in the substantia nigra. Consequently, there is less dopamine in the striatum because substantia nigra neurons project to the striatum. MPTP (1-methyl-4-phenyl-1,2,3,6-tetrahydropyridine) and its active metabolite MPP+ (1-methyl-4-phenylpyridinium), are toxic chemical compounds that cause neurodegeneration of substantia nigra neurons. Therefore, MPTP and MPP+ treated animals are used as models of Parkinson’s disease [31,32].

Different interconnected mechanisms cause substantia nigra neurodegeneration in Parkinson’s disease, for example, alpha-synuclein aggregation, mitochondrial dysfunction, and microglial activation [31,32]. It was found that activated microglia contribute to the neurodegeneration process in the substantia nigra by producing oxidants and proinflammatory cytokines [31]. 

It was repeatedly found that capsaicin reduced neurodegeneration and motor impairment in animal models of Parkinson’s disease [12,31,33,34,35,36]. This compound exerted beneficial effects on Parkinson’s disease by decreasing microglial activation and reducing neuroinflammation [12,31,35,37]. These findings are summarized in Figure 3.

In an MPTP mouse model of Parkinson’s disease, it was found that intraperitoneal application of capsaicin (0.5 mg/kg) increased the number of dopaminergic (tyrosine hydroxylase positive) neurons in the substantia nigra. This effect was dependent on TRPV1 channels. The authors also found that capsaicin improved motor impairment and suggested that one of the mechanisms of these beneficial effects was that this compound decreased the production of reactive oxygen species and proinflammatory cytokines (TNF-α and IL-β) by activated microglia [31]. Similar results were obtained by a different laboratory in a lipopolysaccharide rat model of Parkinson’s disease. The authors found that intraperitoneal capsaicin (1 mg/kg) reduced neurodegeneration in the substantia nigra. In the presence of the tested compound, microglial cells shifted from a proinflammatory to an anti-inflammatory state and produced fewer oxidants and proinflammatory cytokines (IL-1β and IL-6). These neuroprotective effects were dependent on TRPV1 channels because they were reduced after administration of a selective TRPV1 inhibitor, capsazepine [35]. 

Not only microglial cells but also astrocytes (a different type of glial cell) are influenced by capsaicin in Parkinson’s disease [36]. It was found in an MPP+ animal model that capsaicin applied intraperitoneally (1 mg/kg; a single injection/day for 7 days) activated TRPV1 receptors on astrocytes in the substantia nigra. TRPV1 receptors activation enhanced the production of a ciliary neurotrophic factor by astrocytes which increased tyrosine hydroxylase activity in the substantia nigra and dopamine levels in the striatum [36]. Capsaicin also caused behavioral recovery in MPP+ animals. In a different study from the same laboratory, it was found in the same model of Parkinson’s disease that activated microglia produced fewer reactive oxygen species in the substantia nigra after intraperitoneal administration of capsaicin (1 mg/kg). Microglia-derived oxidative stress was reduced by a ciliary neurotrophic factor produced by capsaicin-stimulated astrocytes in the substantia nigra [12]. This effect of capsaicin reduced neurodegeneration and decreased motor impairment in MPP+ rats.

Alpha-synuclein deposition is an important feature of Parkinson’s disease [32]. A protective role of dietary capsaicin (20, 40, 80 and 100 μM for 24 days) against Parkinson’s disease was reported in flies expressing human alpha-synuclein. It was found that capsaicin increased dopamine content, reduced oxidative stress markers, and enhanced free radical scavenging potential in brains obtained from flies with Parkinson’s disease [38].

To summarize, capsaicin treatment reduces neurodegeneration and improves behavioral outcomes in different animal models of Parkinson’s disease. The important mechanism of this effect is that this compound reduces oxidants and proinflammatory cytokines production by activated microglia. This process is shown schematically in Figure 3.

## 5. Effects of Capsaicin in Animal Models of Epilepsy

Epilepsy is the most prevalent chronic neurological disease. The main symptoms are generalized or focal seizures. The backbone of therapy is antiseizure medications which help control symptoms, acting mainly through influencing voltage-gated sodium channels or gamma-aminobutyric acid pathways [39,40,41]. One in four epilepsy cases is resistant to currently available pharmacologic therapy and sadly, drug-resistant epilepsy is more prevalent among children [42]. Numerous efforts are made to design new pharmaceuticals, with 30 drug candidates currently at the preclinical or clinical stage [43], however, their safety profile is yet to be established. 

The literature on capsaicin effects in epilepsy is sparse and, at least at first glance, appears conflicting because both pro- and antiepileptic mechanisms of action are described [44,45,46]. Most of the studies focused on the effects of capsaicin on the TRPV1 channels. Calcium ions are widely believed to play a major role in epilepsy pathogenesis [47,48] and, therefore, TRPV1 channels, with their high permeability to calcium ions, are an attractive target for drug action. Activation of the TRPV1 channel also leads to membrane depolarization and to an increase in glutaminergic activity [49,50]. Moreover, some studies showed that the activation of TRPV1 channels decreases GABA release [51]. All these mechanisms suggest that TRPV1 activation could have proepileptic effects, which has been shown in several studies [45,52]. 

Pentylenetetrazol (PTZ) potently increases neuronal excitability by inhibiting GABA receptors/channels. It was found that injections of this compound evoked seizures in higher percentages of animals after intra-cerebrovascular application of capsaicin (1 or 10 μg/mouse). This effect was abolished by pretreatment with a synthetic TRPV1 antagonist, capsazepine. Moreover, intra-cerebrovascular application of TRPV1 antagonist alone protected against PTZ-induced seizures [52]. Different authors studied the effects of capsaicin in vivo in acute rat models of temporal lobe epilepsy. Epileptic activity was evoked by a stimulating electrode in the hippocampal perforant path. The recording electrode measuring the extracellular potential was placed in the hippocampal formation. The authors reported that intraperitoneal application of capsaicin (10 mg/kg) increased epileptic activity. Capsazepine, applied without capsaicin, showed antiepileptic effects which indicated that TRPV1 receptors were involved [45]. This finding also suggested that in hippocampal neurons, TRPV1 receptors/channels were tonically active under epileptic conditions. 

In a different study, in vitro patch-clamp recordings of excitatory postsynaptic currents (EPSCs) were conducted in hippocampal slices in a proepileptic extracellular solution containing no magnesium and a GABA antagonist—bicuculline. The experiments showed that capsaicin (1–10 μM) enhanced the frequency of EPSCs in slices obtained from mice with pilocarpine-induced epilepsy compared to control animals. The proepileptic action of capsaicin was mediated via increased glutamate release. In the same study, using the Western blotting technique, it was found that the expression of TRPV1 protein was increased in hippocampal neurons obtained from epileptic mice. The authors concluded that capsaicin acted on newly expressed TRPV1 channels in presynaptic axons which enhanced glutamate release in epileptic mice [53]. Similarly, different authors reported an increased expression of TRPV1 receptors in hippocampal pyramidal neurons in rats with pilocarpine-induced epilepsy [54]. 

In another study, capsaicin (10 μM and 100 μM) enhanced epileptiform activity in hippocampal slices in vitro, with epileptiform events evoked in the presence of a potassium channel inhibitor 4-aminopyridine (4-AP). In the same in vitro preparation, TRPV1 antagonist capsazepine applied without capsaicin suppressed epileptic activity. This result suggested a tonic activation of TRPV1 channels in hippocampal neurons under epileptic conditions. The authors also conducted in vivo recordings and obtained similar conclusions. Capsazepine, a TRPV1 antagonist, when applied subcutaneously, inhibited epileptic activity evoked by intrahippocampal administration of 4-AP. The recording electrode was placed in the hippocampus [55].

The above-mentioned studies suggested that TRPV1 channels activation in hippocampal neurons is involved in the proepileptic action of capsaicin. However, TRPV1-independent antiepileptic effects of capsaicin were reported in two in vitro studies in cortical neurons. In our previous publication, we recorded ictal epileptic events lasting more than 100 s using the patch-clamp technique in zero magnesium solution containing 4-AP in prefrontal cortex pyramidal neurons (Figure 4). Additionally, we recorded interictal epileptiform events lasting less than 3 s. Capsaicin (60 μM) potently inhibited both types of epileptic events [3]. We hypothesized that sodium channels, and not TRPV1 channels, were involved in the antiepileptic action of capsaicin, as this compound inhibited voltage-gated sodium channels and action potentials in cortical neurons in our study [3]. Such effects are exerted by many antiepileptic drugs and drug candidates [56,57,58]. Different authors also studied the effects of capsaicin in neocortical pyramidal neurons in vitro. Patch-clamp recordings showed that capsaicin inhibited epileptiform events evoked by applying a proepileptic solution containing GABA receptor antagonist gabazine. The authors also reported that capsaicin (25 μM) reduced the number of action potentials and decreased the maximal amplitude of single-action potential which indicated that the inhibition of voltage-gated sodium currents may have been involved in the antiepileptic action of capsaicin [44]. 

Thus, in the in vitro studies postulating antiepileptic effects of capsaicin, including our study, epileptiform events were induced in cortical neurons and a TRPV1-independent mechanism of action was most likely. On the other hand, the influence of capsaicin on TRPV1 channels expressed in hippocampal neurons consistently showed the proepileptic effects of this compound (Figure 4). Antiepileptic effects of TRPV1 antagonist capsazepine were also described in hippocampal neurons [45,55]. It can be hypothesized that in certain brain areas such as the hippocampus, proepileptic actions of capsaicin prevail whereas, in different brain areas such as the cerebral cortex, antiepileptic actions of capsaicin predominate.

Indeed, it was reported that capsaicin applied subcutaneously (1 mg/kg) decreased behavioral seizures induced by intraperitoneal kainic acid injections in mice [59]. Capsaicin acted on different brain areas because systemic application of kainic acid induces epileptic activity in different brain regions [60]. It may be hypothesized that capsaicin suppressed seizures because the antiepileptic actions of capsaicin were more pronounced than the proepileptic effects [59]. The authors did not conduct experiments aiming to indicate a specific molecular target responsible for the antiepileptic effects of capsaicin. 

To summarize, both proepileptic and antiepileptic effects of capsaicin were described in animal models. This is shown in Figure 4 in simplified form. Studies assessing the effects of dietary capsaicin in human epileptic subjects are, therefore, needed.

## 6. Stroke

Stroke is a common, life-threatening neurological condition. There are two types of stroke: ischemic and hemorrhagic. In the first type, a blood clot forms in an atherosclerotic cerebral artery causing inadequate blood flow and ischemia in the brain area supplied by the blocked artery. Consequently, thrombolytic agents are often used to treat acute ischemic stroke. In the second type of stroke, a certain brain area is damaged because a rupture of a cerebral artery occurs which causes bleeding into the brain. Ischemic stroke is more frequent than hemorrhagic stroke [61]. A low cholesterol diet is essential in preventing ischemic stroke because atherosclerotic plaques that narrow cerebral arteries are mainly composed of cholesterol. There is a variety of symptoms in acute stroke and many serious long-term complications such as motor impairment, speech disorders, and dysphagia. The common animal model of ischemic stroke is cerebral artery occlusion followed by reperfusion [61].

### 6.1. Beneficial Effects of Capsaicin in Animal Models of Stroke

There are a number of publications showing the neuroprotective effect of capsaicin in experimental models of stroke [62,63]. It was found in a middle cerebral artery occlusion/reperfusion model in rats, that capsaicin (1 nmol or 3 nmol) injected into the peri-infarct area decreased infarct volume and improved neurological deficits. The authors conducted cell culture experiments and proposed that capsaicin (3 µM and 10 µM) caused TRPV1 dependent downregulation of NMDA receptors in cortical neurons, which reduced calcium inflow through NMDA receptors and consequently decreased excitotoxicity which contributes to cell death during stroke [62]. It was also found in Mongolian gerbils that capsaicin decreased neuronal cell death caused by brain hypoxia which was induced by internal carotid occlusion for 10 min. Capsaicin (0.01, 0.025, 0.05, 0.2 and 0.6 mg/kg) was injected subcutaneously 5 min after recirculation. The authors hypothesized that capsaicin provided neuroprotection by desensitizing neuronal TRPV1 channels which reduced TRPV1-mediated calcium inflow and decreased excitotoxicity [64].

Capsaicin exerts beneficial effects in stroke models not only by enhancing neuroprotection but also by influencing cerebral vasculature. In one study, the effects of the intraperitoneal application of capsaicin (0.2 mg/kg or 2.0 mg/kg) were assessed in young rats in carotid artery occlusion followed by a global hypoxia stroke model. It was found that capsaicin pretreatment reduced infarct size. The authors suggested that capsaicin treatment caused better oxygenation of brain tissue because myogenic autoregulation of cerebral blood flow was improved after the application of the tested compound [63]. Moreover, it was reported that dietary capsaicin (0.02% in a chow) delays the onset of stroke in stroke-prone rats with hypertension. The authors presented evidence that capsaicin activated TRPV1 receptors/channels expressed in the endothelium of cerebral arteries. Activation of TRPV1 channels increased the activity of endothelial nitric oxide synthase which enhanced the production of relaxing factor nitric oxide. Increased synthesis of nitric oxide enhanced relaxation of the cerebral arteries and prevented stroke [8]. 

### 6.2. Capsaicin Is Helpful in Reducing Stroke Complications in Humans

Dysphagia is a common complication after stroke [65]. It was shown that capsaicin improves the swallowing function in stroke patients presenting clinical signs of dysphagia. Capsaicin (150 μM) was applied to drinking water. Additionally, oropharyngeal mucosa was touched with a swab soaked with capsaicin. The authors speculated that capsaicin activated TRPV1 channels in sensory receptors in the oropharyngeal mucosa which improved sensory input to the swallowing center [65]. In a different study, the swallowing function was compared in two groups of patients with dysphagia after stroke. In the first group, the palate and tongue were stimulated twice a day for several weeks with ice made of normal saline. In the second group of patients, the same stimulation was performed with the exception that the ice was made of saline with capsaicin (150 μM). Swallowing function improved in both groups, but the effect was significantly stronger in the capsaicin-treated patients [66].

In a randomized trial, the effects of oral capsaicin (10 µM) dissolved in tomato juice were compared with transcutaneous sensory electrical stimulation in patients suffering from dysphagia. Both therapies improved swallowing, however, the response rate was higher in patients treated with oral capsaicin. The dysphagia was caused by stroke, advanced age, or other diseases [67]. In a different randomized trial, the same authors assessed the effects of oral capsaicin and other neurorehabilitation strategies on swallow response and motor cortex excitability in patients with post-stroke dysphagia. Although capsaicin (10 µM dissolved in water) had no effect on the biomechanics of swallowing (neither did other interventions), it significantly enhanced pharyngeal and primary motor cortex excitability. Capsaicin had the strongest effects out of all investigated strategies. These results suggest that capsaicin could rapidly induce functional changes in the pharyngeal motor cortex responsible for swallowing. No adverse events of capsaicin were observed [68]. There is currently a phase II clinical CADYS study underway (Capsaicin for Post-stroke Dysphagia) which randomizes stroke patients with dysphagia to receive either a 1% oral solution of capsaicin or a placebo. The assessment of the swallowing function will be performed with standardized tests and the results should be available in 2023 (number of clinical trial NCT04470752). 

The clearance of airway secretions is often impaired after stroke. It was found that nebulized capsaicin (0.49 µM and doubling in dose up to a maximum of 1000 µM) enhanced coughing in a group of non-tracheotomized stroke patients [69]. Moreover, it was reported that capsaicin nebulization (62.5 μM) combined with routine care enhanced sputum excretion in patients tracheotomized after a hemorrhagic stroke. Capsaicin treatment, however, did not significantly improve cough function [70].

Different authors have studied the potential of capsaicin to increase cerebral blood flow, a feature that could be helpful in the setting of ischemic stroke. Capsaicin was tested on healthy volunteers with doses ranging from 33 to 165 μM. During and after the study, there were no side effects. All the tested doses showed the same pattern, which consisted of an increase in the middle cerebral artery mean velocity and a reduction in the middle cerebral artery pulsatility index. This result suggests that capsaicin enhanced cerebral blood flow, which warrants further studies in stroke patients [71].

In summary, it was found in animal stroke models that capsaicin reduced the infarct area and improved neurological deficits. There are few clinical studies that have suggested positive effects of capsaicin in the treatment of stroke complications such as dysphagia. 

## 7. Antidepressant Effects of Capsaicin

The main symptoms of depression are depressed mood, slow thinking, and suicidal tendencies. Several drugs are administered to alleviate the symptoms of this disease such as serotonin reuptake inhibitors. It has also been shown that dietary interventions are helpful in reducing depressive symptoms [14,72]. Immobilization stress or lipopolysaccharide injections are often used as models of depression in animal studies [14,73]. The severity of depressive symptoms and the effectiveness of antidepressive drugs are evaluated in the forced swimming test and tail suspension test [73,74,75]. These tests are used in animal models of depression and in healthy animals because healthy subjects present depressive symptoms under stressful situations such as forced swimming [74].

Several reports have shown the antidepressant effects of capsaicin in animal models. In one interesting study, mice were fed a chow containing capsaicin (0.005% for four months) and, after that, depressive symptoms were evoked by lipopolysaccharide injections. Behavioral tests such as a forced swimming test and tail suspension test showed that depressive symptoms were less pronounced in animals on a capsaicin-rich diet as compared to control. The authors also found that lipopolysaccharide injections changed the composition of gut microbiota and decreased serum serotonin levels. Importantly, these changes were partially reversed by dietary capsaicin [14].

It was reported that intraperitoneal administration of capsaicin produced an antidepressant effect in rats and mice. The authors assessed the antidepressant effects of capsaicin in forced swimming tests [73,74,75]. Interestingly, capsaicin not only exerted antidepressive effects but also enhanced the effectiveness of commonly used antidepressants. It was found that low, subthreshold doses of capsaicin (1 pg/kg, 1 ng/kg, and 0.001 mg/kg, intraperitoneally) combined with a subthreshold dose of amitriptyline induced an antidepressant effect [74]. It was also found that coadministration of capsaicin (0.002 mg/kg, intraperitoneally) and selective serotonin reuptake inhibitor citalopram exerted synergistic antidepressive effects in rats. Moreover, combined administration of the two compounds reduced side-effects of citalopram such as impairment of spatial memory and learning [75]. Different authors found that not only capsaicin (0.1, 1 and 2.5 mg/kg, intraperitoneally) but also a different TRPV1 agonist, olvanil, reduced depressive behavior caused by nicotine treatment or immobilization stress. The authors suggested that TRPV1 channels are involved in the antidepressant effect of capsaicin and olvanil [73]. It was also reported that intracerebroventricular administration of capsaicin (10 μg/mouse) reduced depressive behavior caused by amphetamine withdrawal. The effect was shown to be dependent on TRPV1 channels [76].

Stress may exacerbate symptoms of depression and other psychiatric disorders [77]. Long-term potentiation (LTP) is activity-dependent synaptic plasticity which is an electrophysiological correlate of certain forms of memory such as spatial memory [24,25,77]. It was found that acute stress evoked by placing animals on an elevated platform suppressed LTP and that this effect was prevented by capsaicin. LTP was recorded in vitro in hippocampal slices obtained from young, stressed rats. Capsaicin (1 mM) was applied in vitro before LTP induction. In the same report, it was found that acute stress impaired spatial memory retrieval. This negative effect was prevented by intrahippocampal infusion of capsaicin (1 mM). Thus, in vitro electrophysiological experiments correlated with behavioral experiments. The beneficial effects of capsaicin were mediated by TRPV1 channels [77]. The authors suggested that activating TRPV1 channels may reverse stress-evoked spatial memory impairment.

Other reports, however, have suggested that inhibition of TRPV1 channels reduced the effects of stress which is in contrast to the study presented above. For example, it was found that injections of the TRPV1 antagonist capsazepine to midbrain periaqueductal grey matter reduced behavioral effects of stress in rats exposed to a predator [78].

In essence, capsaicin reduces depressive behaviors in animal models as shown by behavioral tests. Moreover, studies show that capsaicin may enhance the anti-depressive effects of commonly used antidepressants. 

## 8. Beneficial Effects of Capsaicin in the Treatment of Headaches

Migraine is a common disorder that is characterized by disabling headaches and many other symptoms such as increased sensitivity to light and psychiatric disturbances. A migraine is either episodic or chronic [79]. A cluster headache is a different type of headache that is shorter in duration than a migraine attack [79,80,81]. It was found that calcitonin gene-related peptide (CGRP), a potent vasodilator, contributes to the pathogenesis of migraine and cluster headaches. This peptide is released from trigeminal fibers innervating cerebral vasculature and causes the dilation of cerebral blood vessels which evokes pain. Consequently, CGRP and CGRP receptors are new therapeutic targets for drugs used in the treatment of migraine and cluster headaches [79]. The beneficial effects of capsaicin in these two types of headaches were previously described [82,83,84,85].

People suffering from migraines may experience pain at pressure points on the scalp arteries between attacks [85]. In a small clinical study, it was found that capsaicin jelly (0.1%) applied to painful arteries reduced their tenderness in the interictal period in the majority of patients. Moreover, the same method of capsaicin application reduced the severity of migraine attacks [85]. It was also shown that intranasal application of capsaicin (10 mM) reduced pain intensity in chronic migraine patients compared to placebo treatment. Burning sensations evoked by capsaicin were well-tolerated. Moreover, they decreased after each consecutive capsaicin application because the effect was desensitized. In placebo-treated patients, burning sensations of decreasing intensity were evoked by repeated intranasal application of citric acid solution (its pH was increased after each application) [82].

It was also shown that intranasal capsaicin (10 mM) application exerted pain-relieving effects in cluster headache patients. The authors reported that intranasal application of capsaicin on the painful side of the head (ipsilateral side) gave better results than capsaicin application on the contralateral side [80].

In a different study, it was found that intranasal application of capsaicin (10 mM) in cluster headache patients reduced the frequency of pain attacks. Cerebral circulation was likely involved in this effect because middle cerebral arteries were narrowed in a group of healthy humans after a single application of capsaicin, measured with a Doppler device. In the same report, it was shown in animal studies that intranasal capsaicin reduced the immunoreactivity to substance P and CGRP in sensory fibers innervating nasal mucosa. The authors suggested that capsaicin may exert the same effect in cluster headache patients, which could contribute to the pain-relieving effects of the tested compound [81].

To summarize, few clinical studies show pain-relieving effects of topical capsaicin in migraine and cluster headaches. One of the limitations of these studies, however, was the small number of patients involved. 

## 9. Concluding Remarks

Several recent studies performed on animals assessed the effects of capsaicin in brain disorders using histological, behavioral, genetic, and electrophysiological techniques. Most publications described the effects of capsaicin in neurodegenerative diseases, stroke, and epilepsy. It was reported that capsaicin decreased neurodegeneration in Alzheimer’s and Parkinson’s diseases. Moreover, it was found that capsaicin pretreatment reduced the area of infarction in experimental models of stroke. In the case of epilepsy, both beneficial and adverse effects were described. The tested compound exerted proepileptic effects by opening TRPV1 channels, and antiepileptic effects by inhibiting voltage-gated sodium channels. Capsaicin reduced symptoms of depression in animal models. There are also human studies suggesting the analgesic effects of topical capsaicin in different kinds of headaches. Capsaicin was helpful in treating post-stroke dysphagia. Moreover, nebulized capsaicin enhanced airway clearance in stroke patients. 

Capsaicin is a safe compound because it is often consumed in different countries. Spices containing capsaicin are relatively cheap. We suggest that chili peppers should be added to meals regularly. Considering that, besides capsaicin, they contain precursors of vitamin A, vitamin C, and other antioxidants [86]. Chili peppers may be helpful not only in neurological but also in cardiovascular and oncological diseases [6,7,12,13]. Chili peppers may also help obese people to lose weight [5,86]. People with epilepsy should eat them with caution, however, because some studies have shown that capsaicin may enhance epileptic symptoms. 

Without a doubt, the health effects of capsaicin are mostly positive. Further preclinical research aiming to elucidate the effects of capsaicin in brain disorders is required. For example, modifying the chemical structure of capsaicin may improve the pharmacological properties of this compound. More studies involving larger groups of patients are needed to assess the full therapeutic potential of capsaicin in brain disorders.

## Figures and Tables

**Figure 1 molecules-27-02484-f001:**
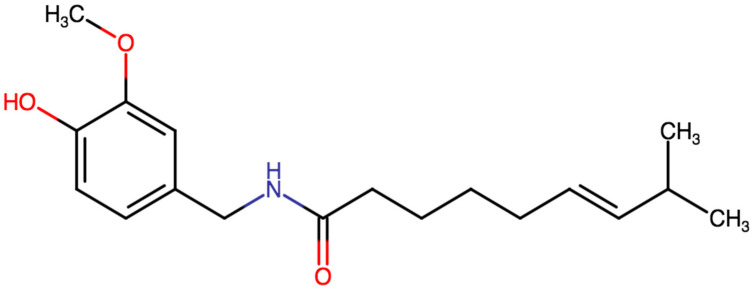
Chemical structure of capsaicin.

**Figure 2 molecules-27-02484-f002:**
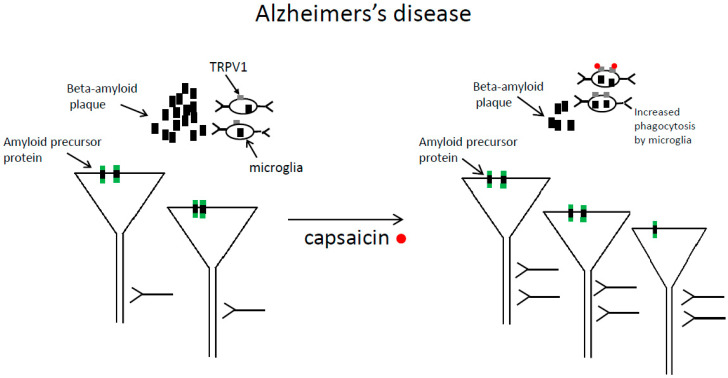
The beneficial effects of capsaicin in Alzheimer’s disease are shown in simplified form. Capsaicin treatment increases the number of cortical pyramidal neurons and reduces synapse loss. Additionally, capsaicin decreases beta-amyloid deposition in the extracellular space. One of the mechanisms is that capsaicin treatment enhances phagocytosis of beta-amyloid plaques by microglial cells (capsaicin binds TRPV1 receptors expressed on microglial cells). Left panel—before capsaicin treatment, right panel—after capsaicin treatment.

**Figure 3 molecules-27-02484-f003:**
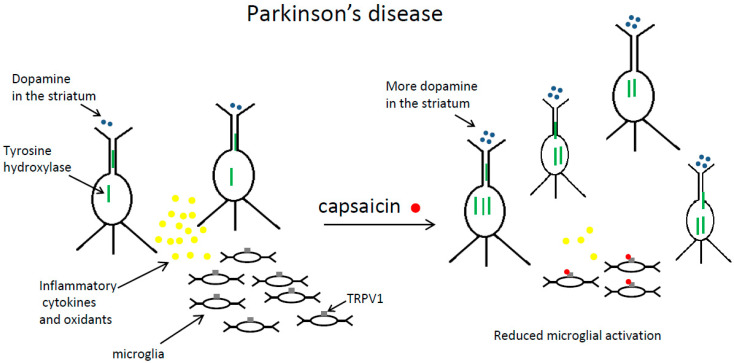
The beneficial effects of capsaicin in Parkinson’s disease are shown in simplified form. Capsaicin administration increases the number of dopaminergic neurons in the substantia nigra and the expression of tyrosine hydroxylase in these neurons. Consequently, there is more dopamine in the striatum. Moreover, after capsaicin treatment, microglial cells in the substantia nigra produce fewer oxidants and inflammatory cytokines. Left panel—before capsaicin treatment, right panel—after capsaicin treatment.

**Figure 4 molecules-27-02484-f004:**
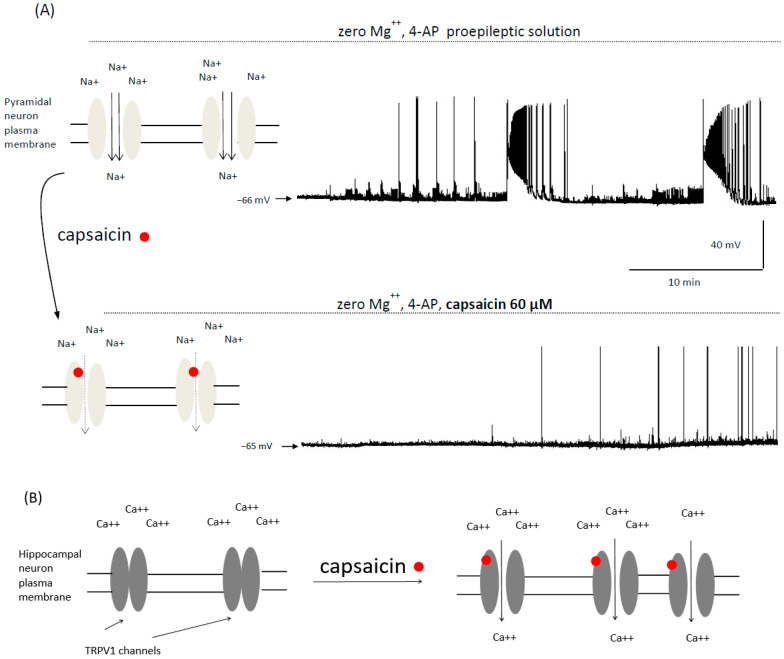
Animal studies show that capsaicin exerts both anti- and proepileptic effects. (**A**)—the top panel shows a schematic drawing of voltage-gated sodium channels in the plasma membrane of a brain cortical neuron. Example epileptic events recorded from a cortical neuron in vitro are also shown. A patch-clamp recording was made in proepileptic extracellular solution containing zero magnesium and potassium channels inhibitor 4-AP. The bottom panel shows that capsaicin potently inhibits epileptic activity [3]. The blockading of sodium channels by capsaicin removes depolarization and contributes to the antiepileptic effect of capsaicin. Please note that a number of other channels contribute to the generation of epileptic seizures such as calcium channels which may also be blocked by capsaicin. The recordings shown in Figure 4A were presented in our previous publication [3]. (**B**)—schematic drawing showing that capsaicin exerts proepileptic effects by the opening of calcium-permeable TRPV1 receptor/channels in hippocampal neurons [53,55]. Some of these channels are newly expressed under epileptic conditions. Calcium inflow causes depolarization of the membrane potential and exerts a proepileptic effect. If this process occurs in presynaptic axons, glutamate release is enhanced which may evoke seizures.

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
