# Peer review of "Beneficial Effects of Capsaicin in Disorders of the Central Nervous System"

_molecules, 2022, doi:10.3390/molecules27082484_

Round 1
Reviewer 1 Report
The present manuscript by Michał Pasierski and Bartłomiej Szulczyk reviews effects of capsaicin in central nervous system disorders.
Here, in the Abstract, the authors mention a list of beneficial effects of capsaicin and its membrane protein targets: TRPV1 channels and voltage-gated sodium channels.
Activation of TRPV1 channels by capsaicin (a compound found in chili peppers) in nociceptive sensory neurons contributes to its analgesic effect. Other beneficial effects of capsaicin have been described in cardiovascular and oncological disorders, Alzheimer’s, and Parkinson’s diseases, depression in animal models of stroke, in migraine and cluster headaches, whereas opposite effects of capsaicin in animal models of epilepsy.
The theme is interesting and clinically relevant. The manuscript could be improved in some points.
Comments to the authors:
References: avoid citing references with both numerals and authors throughout the manuscript.
It would be beneficial for the reader put the references in numerical order.
Line 187: 12, 28, 30-33 instead of 30, 31, 28, 32, 33, 12.
Line 190: 12, 28, 32, 34 instead of 34, 12, 28, 32.
Fig. 1: What does APP mean?
Fig. 2: As in Fig. 1, illustrate the TRPV1 receptor (the target for capsaicin) in microglia cells.
Fig. 3: To better understand the effect of capsaicin on the membrane potential, it would be adequate to indicate the level of resting membrane potential close to the records.
In several parts throughout the manuscript, the text in the same paragraph refers to only one article. This paper is cited two or three times. The information therein sometimes is redundant, it could be reduced. To mention two examples, see lines 155-168 and 213-219.
Line 177: What do MPTP and its active metabolite MPP+ mean?
Lines 530, 531, delete complete name: calcitonin gene-related peptide, which was mentioned in lines 501, 502.
The paragraphs in the "Concluding remarks" are very general. This section needs more concise conclusions.
Although most of the effects reported for capsaicin are beneficial, in the context of the present manuscript, it is interesting the paper “TRPV1 blockers as potential new treatments for psychiatric disorders by Iglesias et al., Behav Pharmacol. 2022; 33:2-14“, in which TRPV1 blockers are effective in psychiatric disorders such as depression, anxiety, panic, conditioned fear, drug addiction; in comparison with the effects an agonist of TRPV1 receptors (capsaicin), in central nervous system disorders reviewed here. It would be constructive to include this material to discussing other contrast findings.
What the authors could mention regarding persons that include chili peppers in their diet as beneficial or harmful effects? A brief paragraph could be included. See for instance Antioxidant, Anti-Obesity, Nutritional and Other Beneficial Effects of Different Chili Pepper, by Azlan et al., Molecules. 2022; 27(3):898.
Author Response
Reviewer 1
The present manuscript by Michał Pasierski and Bartłomiej Szulczyk reviews effects of capsaicin in central nervous system disorders. Here, in the Abstract, the authors mention a list of beneficial effects of capsaicin and its membrane protein targets: TRPV1 channels and voltage-gated sodium channels. Activation of TRPV1 channels by capsaicin (a compound found in chili peppers) in nociceptive sensory neurons contributes to its analgesic effect. Other beneficial effects of capsaicin have been described in cardiovascular and oncological disorders, Alzheimer’s, and Parkinson’s diseases, depression in animal models of stroke, in migraine and cluster headaches, whereas opposite effects of capsaicin in animal models of epilepsy. The theme is interesting and clinically relevant. The manuscript could be improved in some points.
Comments to the authors:
- References: avoid citing references with both numerals and authors throughout the manuscript. It would be beneficial for the reader put the references in numerical order.
Line 187: 12, 28, 30-33 instead of 30, 31, 28, 32, 33, 12.
Line 190: 12, 28, 32, 34 instead of 34, 12, 28, 32.
All citations have been corrected, as suggested by the reviewer (see revised manuscript).
- Fig. 1: What does APP mean?
It has been explained that APP means amyloid precursor protein.
- Fig. 2: As in Fig. 1, illustrate the TRPV1 receptor (the target for capsaicin) in microglia cells.
TRPV1 receptors in microglial cells have been illustrated.
- Fig. 3: To better understand the effect of capsaicin on the membrane potential, it would be adequate to indicate the level of resting membrane potential close to the records.
The values of the membrane potential have been indicated in the Figure.
- In several parts throughout the manuscript, the text in the same paragraph refers to only one article. This paper is cited two or three times. The information therein sometimes is redundant, it could be reduced. To mention two examples, see lines 155-168 and 213-219.
We removed redundant citations of the same article in lines indicated by the reviewer. We did the same in other parts of the text (see revised manuscript).
- Line 177: What do MPTP and its active metabolite MPP+ mean?
Abbreviations MPTP and MPP+ have been defined (see first lines of subsection 4).
- Lines 530, 531, delete complete name: calcitonin gene-related peptide, which was mentioned in lines 501, 502.
This has been done.
- The paragraphs in the "Concluding remarks" are very general. This section needs more concise conclusions.
Concluding remarks (Subsection 9) have been rewritten and extended (see revised manuscript, marked in red).
- Although most of the effects reported for capsaicin are beneficial, in the context of the present manuscript, it is interesting the paper “TRPV1 blockers as potential new treatments for psychiatric disorders by Iglesias et al., Behav Pharmacol. 2022; 33:2-14“, in which TRPV1 blockers are effective in psychiatric disorders such as depression, anxiety, panic, conditioned fear, drug addiction; in comparison with the effects an agonist of TRPV1 receptors (capsaicin), in central nervous system disorders reviewed here. It would be constructive to include this material to discussing other contrast findings.
The publication mentioned by the reviewer has now been cited in our manuscript as one of the findings which are in contrast to many reports presenting beneficial effects of capsaicin (see last lines of subsection 7, marked in red).
- What the authors could mention regarding persons that include chili peppers in their diet as beneficial or harmful effects? A brief paragraph could be included. See for instance Antioxidant, Anti-Obesity, Nutritional and Other Beneficial Effects of Different Chili Pepper, by Azlan et al., Molecules. 2022; 27(3):898.
In the conclusion section, more advice have been given for people who include chili peppers in their diet (second paragraph of the revised conclusions, marked in red).
The manuscript has been edited by a native speaker.
Reviewer 2 Report
molecules-1641008, Beneficial effects of capsaicin in central nervous system disorders
The manuscript presents a literature review on capsaicin and its effects targeted at central nervous system disorders. Overall the work in organized and could be of interest for the journal’s readers. There are some problems and shortages that need the authors’ attentions.
Considering the profile of the journal, please add a structure of capsaicin. The authors could also add some physico-chemical properties, like logP, solubility, and some pharmacokinetics properties: bonding on proteins, half-life, brain blood distribution, and others that a significant.
An important problem is the lack of information on the doses. The authors should check all the manuscript and add the doses used in each experiment described. The readers should be able to understand if the described effect needs a low, a medium, or a high dose of capsaicin.
There are close to 300 clinical studies on capsaicin described in the portal www.clinicaltrials.gov. The authors should analyze all the studies focus on central nervous system disorders. A new section should be added to describe the clinical data on capsaicin. Please highlight the efficacy or the lack of it and the side-effects observed. This section should be most important.
There are many editing mistakes. See the reference style in the text, see the presence of multiple consecutive spaces, and several others.
The authors describe in some sections some derivatives of capsaicin, like dihydrocapsaicin. It is not relevant and I advise the authors to remove those.
Row 548, what does it mean cheap? In my opinion, capsaicin is not cheap. The purity of the compound is a major factor for the price.
Author Response
Reviewer 2
The manuscript presents a literature review on capsaicin and its effects targeted at central nervous system disorders. Overall the work in organized and could be of interest for the journal’s readers. There are some problems and shortages that need the authors’ attentions.
- Considering the profile of the journal, please add a structure of capsaicin. The authors could also add some physico-chemical properties, like logP, solubility, and some pharmacokinetics properties: bonding on proteins, half-life, brain blood distribution, and others that a significant.
Additional Figure has been added showing the chemical structure of capsaicin (Figure 1). Moreover, some physico-chemical and pharmacokinetic properties of capsaicin have been described (subsection 2, marked in red).
- An important problem is the lack of information on the doses. The authors should check all the manuscript and add the doses used in each experiment described. The readers should be able to understand if the described effect needs a low, a medium, or a high dose of capsaicin.
Doses of capsaicin have been indicated throughout the text (see revised manuscript, marked in red).
- There are close to 300 clinical studies on capsaicin described in the portal www.clinicaltrials.gov. The authors should analyze all the studies focus on central nervous system disorders. A new section should be added to describe the clinical data on capsaicin. Please highlight the efficacy or the lack of it and the side-effects observed. This section should be most important.
We thank the Reviewer for this remark. However, out of 300 studies describing the effects of capsaicin only 13 concern CNS diseases. In 9 studies, capsaicin was used as a methodological tool (for example as a cough stimulant) and not as a therapeutic option. Remaining 4 clinical studies have been described in the revised manuscript. 3 of them have been published (Ortega et al. 2016, Cabib et al. 2020, Marquez-Romero et al. 2021) and one clinical trial has not been finished (NCT04470752). As all four studies deal with stroke, they have been included in the subsection which is about stroke complications in humans (subsection 6.2 second and fourth paragraph, marked in red).
- There are many editing mistakes. See the reference style in the text, see the presence of multiple consecutive spaces, and several others.
Reference style in the text has been corrected. In the revised manuscript, citations are presented as numbers only (authors’ names have been removed). Redundant spaces have been removed. The manuscript has been edited by a native speaker.
- The authors describe in some sections some derivatives of capsaicin, like dihydrocapsaicin. It is not relevant and I advise the authors to remove those.
Citations describing the effects of dihydrocapsaicin have been removed.
- Row 548, what does it mean cheap? In my opinion, capsaicin is not cheap. The purity of the compound is a major factor for the price.
We meant that spices containing capsaicin are cheap. In the revised manuscript, it is now stated that spices containing capsaicin are relatively cheap (subsection 9, second paragraph, marked in red).
Round 2
Reviewer 2 Report
The authors modified the paper based on the review's comments and improved the quality of the manuscript. It still needs a careful check for editing and English mistakes.